# An Evaluation of a Virtual Food Safety Program for Low-Income Families: Applying the Theory of Planned Behavior

**DOI:** 10.3390/foods11030355

**Published:** 2022-01-26

**Authors:** Juan C. Archila-Godínez, Han Chen, Leah Klinestiver, Lia Rosa, Tressie Barrett, Shauna C. Henley, Yaohua Feng

**Affiliations:** 1Department of Food Science, Purdue University, 745 Agriculture Mall Drive, West Lafayette, IN 47907, USA; jarchila@purdue.edu (J.C.A.-G.); chen2401@purdue.edu (H.C.); l_graciel.16@hotmail.com (L.R.); tebarrett@hotmail.com (T.B.); 2Department of Public Health, Purdue University, 610 Purdue Mall, West Lafayette, IN 47907, USA; lklinest@purdue.edu; 3Department of Extension, University of Maryland Extension, 1114 Shawan Road, Baltimore County, MD 21030, USA; shenley@umd.edu

**Keywords:** evaluation, food safety, low-income, theory of planned behavior, virtual

## Abstract

Low-income families are reported to have a limited knowledge of food safety and resources to follow food safety practices compared with the rest of the population. This paper evaluated a virtual food safety educational program targeting food handlers in low-income families. Trained native speakers of English and Spanish delivered course materials in both languages. A total of 60 individuals participated in the program, with 30 participants in each language group. Most were female, and most had fewer than three children. After the program, participants’ food safety knowledge and self-reported safe food practice behavior scores increased significantly from 5.32 to 7.43 (out of 8.00) and from 24.78 to 29.30 (out of 35.00), respectively. The theory of planned behavior (TPB) was used to understand individuals’ behavior change intention of food safety practices. All the TPB constructs’ scores, including attitudes toward the behavior, perceived behavioral control, subjective norms, and behavior change intentions, were improved significantly; however, only the subjective norms and perceived behavioral control were significantly correlated with the behavior change intentions. This virtual educational program improved low-income individuals’ food safety knowledge and changed their food safety attitudes and behaviors, giving a path to develop and evaluate more virtual food safety educational programs in the future.

## 1. Introduction

Foodborne illness is a major public health concern in the United States. An estimated 48 million (one in six) Americans experience foodborne illness every year, 3000 of whom die [1]. As with most health issues, some population groups are more susceptible to contracting a foodborne illness, including children under age five, adults 65 years and older, pregnant women, and the immunocompromised [2]. Furthermore, social determinants, such as an individual’s economic stability, can increase the risk of acquiring a foodborne illness. Therefore, low-income individuals are more susceptible to this condition [3].

Low-income individuals, consisting of a significant portion of minorities in the United States [4], may have a heightened risk due to the unique barriers they face, including the lack of knowledge of some food safety practices, neighborhood geographical factors, cultural factors, and the lack of food handling tools. For example, geospatial data from Detroit, Michigan, suggests that residents of poorer neighborhoods must travel farther to the nearest supermarket than residents of more affluent neighborhoods [5]. Henley et al. [6] reported that minority consumers who rely heavily on public transportation reported long commutes to their preferred grocery store. Moreover, a study investigating the variability of food quality in markets by neighborhoods’ socioeconomic status levels reported higher microbial loads of foodborne pathogens on produce and food-contact surfaces in grocery stores located in low-income neighborhoods [7]. Darcey and Quinlan [8] used a geographic information system to track the food service critical health code violations in Philadelphia and found that more establishments in higher poverty rate areas had at least one health code violation compared to lower poverty rate areas. Additionally, minority groups increase the consumption of a variety of dishes in the United States. Different food cuisine, ethnic food, liked by those minority groups may create unique barriers as some foods, such as soft cheese (e.g., queso fresco) and chitterlings, are linked to higher risk for foodborne illness [9,10]. Another challenge for low-income individuals is the lack of access to food handling tools. For example, one tool, a cooking thermometer, might not be owned by low-income individuals relying on subjective cues like the color of meat when determining its doneness [11,12,13]. Asian and Hispanic consumers were reported to commonly prepare meats in small pieces, which can be cooked in a short period of time that might be insufficient to kill pathogens [6]. Additionally, low-income individuals may own only one cutting board [13], which increases the risk of cross-contamination.

Some of these challenges have been addressed by food safety programs and campaigns for consumers [14,15], but little is known about programs specifically targeting food handlers in low-income families. While these programs and campaigns have been very useful in addressing food safety challenges, they cannot always attack those challenges directly, which means that limited resources can be provided to the participants due to the high cost of educational programs. However, food safety interventions deliver knowledge and skills that can indirectly address some challenges, helping individuals to “think out of the box” and find innovative solutions to address their food safety barriers. Some of the knowledge delivered in these educational programs included basic hygiene, such as handwashing [16], and the core four food safety practices: clean, separate, cook, and chill [17]. Different formats have also been used to deliver these topics to consumers, such as campaigns [17,18], online materials [19], and combinations of formats, such as videos and brochures [20], fotonovela brochures, videos, an animation, a website, and promotional pens containing the website link [21]. The evaluation tools to measure program effectiveness have relied on pre- and post-surveys, with an emphasis on knowledge acquired by the consumers. However, some interventions evaluated other factors: attitudes, perceived risk, and social norms [19].

When evaluating food safety educational programs for food handlers in low-income families, it is critical to measure behavior change and contributing factors to such behavior change. Theories of behavior change, such as the theory of planned behavior (TPB), help understand behaviors among a target population. The TPB postulates that attitudes toward a behavior, perceived behavioral control (PBC), and subjective norms can drive behavior change intentions [22,23]. Attitudes toward the behavior refer to an individual’s perception of anticipated events if a behavior is performed. Subjective norms include the attitudes of members of an individual’s social group toward a particular behavior and the individual’s attitudes toward conforming to those norms. PBC includes perceptions about how much control an individual feels over the behavior change. Behavior change intentions refers to an individual’s plans to perform or not perform a particular behavior [22].

The TPB was used in this study because subjective norms and PBC may help identify food handling behavioral characteristics unique to low-income individuals due to these elements’ ability to reflect social circumstances and economic barriers. The TPB has been used in studies on predicting food safety practices among school food handlers [24], primary preparers of home meals [25], and adolescent populations [26,27]. Few studies have evaluated food safety educational programs developed for low-income individuals using the TPB, despite these populations displaying unique social factors that can influence health attitudes [28].

A dialogue-based learning approach was incorporated in the development of this educational program. It is an interactive learning approach in which the facilitator uses a series of topic-related questions to guide learners in understanding the topic through reflecting their related personal experiences and discussing the questions with their peer learners and the facilitator [29,30]. This approach enables the peer-learning process among learners to learn from the successful experiences of those with similar backgrounds [31]. In the meantime, the facilitator can gain a deeper understanding of learners’ challenges and needs and provide individualized feedback and support accordingly [32].

The present study aimed to evaluate the effectiveness of a dialogue-based virtual-delivery food safety educational program—developed for food handlers in low-income families with young children—in improving participants’ knowledge, attitudes toward the behavior, and behavior change intentions, and to better understand how subjective norms and PBC influence such behavior change.

## 2. Materials and Methods

During the COVID-19 pandemic beginning in 2020, many in-person programs were cut or canceled. To address low-income individuals’ unique barriers to safe food handling in both direct and indirect ways (Appendix A), the Purdue Food Safety Human Factor Team (https://ag.purdue.edu/foodsci/Fenglab/ [accessed on 15 December 2021]) developed a virtual food safety educational program. The educational program included a total of two hours of instructions and two take-home tasks. The two-hour instructions were composed of two weekly facilitator-led course sessions. To ensure the consistency of the delivery, pre-recorded slides were developed in both English and Spanish. The facilitators led the discussion and used the pre-recorded slides to demonstrate food safety concepts. All the learning materials were developed based on the core four food safety practices created by the Fight BAC! campaign (www.fightbac.org [accessed on 15 December 2021]): clean, separate, cook, and chill. The program content was delivered over two course sessions. The first session introduced the concepts of “clean” and “separate”, and the second session introduced “cook” and “chill”. A dialogue-based learning approach was incorporated in the course content delivery. The development of the intervention can be found in Chen et al. [29]. All course materials can be found online: https://ag.purdue.edu/foodsci/Fenglab/extension-articles/ (accessed on 15 December 2021).

The virtual food safety educational program evaluation was composed of a pre- and post-survey. This research protocol #1903021852 was approved by the Institutional Review Board (IRB) at Purdue University on 4 April 2019.

### 2.1. Survey Design

A pre- and post-survey evaluated the effectiveness of this educational program. The TPB was used in the survey design to investigate behavior change intentions of participants. Six constructs were measured in the pre- and post-survey (Appendix A): (1) knowledge, (2) attitudes toward the behavior, (3) PBC, (4) subjective norms, (5) behavior change intentions, and (6) self-reported safe food practice behaviors. All the constructs, except the knowledge construct, were measured in a five-point Likert scale.

The knowledge constructs were adapted from Barrett et al. [17], and the TPB constructs were developed based on Ajzen’s guidelines for developing the TPB questionnaire [33]. The knowledge section included eight multiple-choice questions evaluating participants’ knowledge of (a) recommended handwashing, (b) chilling, and (c) storage practices, as well as (d) recommended meat cooking temperatures and refrigerator temperatures.The attitudes toward the behavior construct included seven questions measuring participants’ attitudes toward the core four food safety practices (clean, separate, cook, and chill), the importance of participants’ children washing their hands, and the risk of their children contracting foodborne illness.The PBC construct contained seven questions measuring participants’ confidence in their ability to comply with the core four food safety practices, ensure their children washed their hands, and prepare safe foods for their children.The subjective norms construct had eight questions measuring the perceived beliefs of participants’ family and friends on the core four food safety practices, the importance of ensuring the handwashing practices of participants’ children, participants’ children’s risk of contracting foodborne illnesses, and the ability of participants to prepare safe foods for their children.The behavior change intentions construct included six questions evaluating participants’ intentions to adopt the core four food safety practices, ensure participants’ children washed their hands, and follow recommended practices to prepare safe foods for their children.The self-reported safe food practice behaviors had seven questions evaluating participants’ current compliance with the core four food safety practices and ensuring participants’ children washed their hands.

The Cronbach’s alpha value of the Likert scale questions was examined for internal consistency. The overall Cronbach’s alpha value of the TPB construct was 0.85, indicating acceptable reliability [34].

In the pre-survey, participants’ sociodemographic information was also collected, including gender, age, ethnicity, number of children in the household, meal preparation frequency, household income, and education level. In the post-survey, six additional program evaluation questions were added to collect participants’ feedback. Surveys (pre- and post-) were reviewed by two food safety experts for content validity and were revised before distribution. The pre-survey had 45 questions, and the post-survey had 51 questions. Both surveys were distributed online using Qualtrics (Provo, UT); each survey took approximately 10 min to complete.

Both the course materials and the pre- and post-surveys were translated into Spanish by a trained bilingual researcher and were verified by another trained bilingual researcher. All collected Spanish information was translated back into English and verified by the same researchers.

### 2.2. Participant Recruitment

Both English and Spanish flyers were sent to and distributed by the local extension offices, local health departments, Women, Infants, and Children (WIC) Program, Food Pantry, and other related community organizations. Some participants were also recruited by word-of-mouth. Those interested in participating were asked to complete a short Qualtrics questionnaire that collected sociodemographic, contact, and availability information. Study eligibility requirements were the following: older than age 18, served as the primary home food handlers, had young children (<5 years old), and were considered low-income, very low-income, or extremely low-income based on the U.S. Department of Housing and Urban Development’s 2019 Section 8 income limit values [35]. In addition, the sample size for this study was determined based on previous food safety intervention programs [26,36,37], and an appropriate number of participants needed to perform statistical analysis of pre- and post-data collected [38,39].

### 2.3. Procedure

The study was conducted between April and October 2020. Due to the COVID-19 pandemic, all participants were recruited and contacted virtually. Participants meeting the selection criteria were contacted via email or phone to confirm their participation in the study and their mailing addresses. After confirmation, program packages with all learning materials, cooking thermometers, and refrigerator thermometers were mailed to the participants, which usually took three to five days to arrive, depending on participant location. The estimate cost of the items in each package was USD 50. Participants were grouped based on program language (English or Spanish) and their availability. English speakers and Spanish speakers received educational programs in their native language. One trained native English-speaking instructor led the English version of the program, while a native Spanish-speaking instructor led the Spanish version. All participants received a total of two weekly one-hour virtual course sessions. Information on how to use and access Zoom was provided to the participants in the days leading up to their first scheduled session. They were required to complete the pre-survey before the beginning of the first session. During the sessions, instructors used the pre-recorded slides to demonstrate the concepts and facilitated the group discussion and the in-class activities. At the end of the second session, all participants were asked to complete the post-survey. Upon the completion of the two course sessions, participants were asked to scan or take photos of their in-class activities and take-home tasks and then to send them to the program instructor via email for further data analysis [29]. All participants received a USD 50 monetary incentive. All course sessions were audio- and video-recorded on Zoom with the permission of study participants.

### 2.4. Data Analysis

The pre- and post-survey were downloaded from Qualtrics and imported into IBM SPSS Statistics (Version 26, SPSS Inc., Chicago, IL, USA) for further data analysis. In the present study, a chi-square test compared the differences of the correct answer frequencies of each knowledge question between the pre- and post-survey. A paired sample t-test was used to determine the mean score change among each item of knowledge, attitudes toward the behavior, PBC, subjective norms, behavior change intentions, self-reported safe food practice behaviors, and to compare the overall mean score. The maximum scores for knowledge, attitudes toward the behavior, PBC, subjective norms, behavior change intentions, and self-reported safe food practice behaviors were 8, 35, 35, 40, 30, and 35 points, respectively. The differences between the overall pre- and post-survey scores were calculated in SPSS, and the means were reported. A Spearman correlation was conducted to determine the strength of association between each factor measured; Table 1 shows the coefficient interpretation adapted from Akoglu [40].

## 3. Results and Discussion

### 3.1. Sociodemographic Characteristics

A total of 60 participants completed the virtual food safety program (participating in both sessions) with an equal number of English (*n* = 30) and Spanish (*n* = 30) participants. Most participants were female (97%), aged 35 to 54 (47%), and prepared meals at home nearly all the time (89%). The high percentage of females matches previous research showing that females are still the primary meal preparers at home [41]. Flagg et al. [42] also found that females were more likely than men to be the primary planners, preparers, and grocery shoppers for their households. A qualitative study found that men whose jobs usually involve physical work were less willing to help in domestic chores than those men whose jobs do not demand physical work [43]. In this study, most participants (80%) also had fewer than three children at home. Participant ethnicity varied depending on the program language. The majority of Spanish participants were Hispanic (77%), while most English participants were White (47%) and African American (34%) (Table 2).

### 3.2. Theory of Planned Behavior

This study used the TPB to measure the intentions of food handlers in low-income families to adopt food safety practices. As shown in Table 3, the overall mean scores of the three constructs measured in the TPB were increased significantly from the pre- to the post-survey, including the attitudes toward the behavior increasing 2.90 points (*p* < 0.001) with a possible total score of 35 points, PBC increasing 2.80 points (*p* < 0.001) with a possible total score of 35 points, and subjective norms increasing 2.33 points (*p* < 0.001) with a possible total score of 40 points. The mean score of the behavior change intentions also increased by 1.75 points (*p* < 0.001) with a possible total score of 30 points, and the self-reported safe food practice behaviors increased by 4.52 points (*p* < 0.001) with a possible total score of 35 points.

The different constructs measured in this study had a significant increase after the educational program. The change in the constructs could be a result of how the program was designed and delivered to the participants. The significant increase in the attitudes toward the behavior corresponds with a stronger belief that the food safety practices discussed in the program can make them and their family safe if they are correctly followed. Other studies have also shown that food handlers, both households and commercial, have reported stronger attitudes about food safety when they have been exposed to food safety topics [44]. The significant increase in PBC corresponds to participants’ stronger confidence in their abilities to perform the food safety practices discussed in the program. In this study, take-home tasks were assigned to participants. Providing hands-on experience can promote an engagement in participants’ food safety behaviors because those activities are linked to daily food preparation at home [45,46]. The significant increase in subjective norms corresponds to participants’ higher belief that their family and friends consider food safety practices important. Humans are social beings and are more likely to follow recommended practices if they know that other people are following them, especially if those people are their peers or family members [47]. The significant increase in the behavior change intentions corresponds to a stronger willingness to perform food safety practices on a daily basis. Finally, the significant increase in the self-reported safe food practice behaviors corresponds to an increase in the performance of the food safety practices discussed in this program. The last two concepts, behavior change intentions and self-reported safe food practice behaviors, can elucidate how the attitudes toward the behavior, PBC, and subjective norms can influence participants’ willingness to adopt food safety practices and to use them in their daily food preparation [22].

Previous studies also utilized the TPB constructs to evaluate the food safety behavior change intentions after educational interventions. Milton and Mullan [26] showed a significant increase in participants’ safe food handling behavior after receiving an intervention; however, compared to the current study, among the TPB constructs, their PBC was the only construct that increased significantly. Mullan and Wong [48] found that more information to emphasize PBC in safe food handling resulted in a significant increase in the PBC construct. A longitudinal study (4 weeks and 12 weeks) by Yardley et al. [16] related to a virtual handwashing intervention showed that handwashing intentions increased after the intervention, and these intentions were higher than those from the control group. Among the TPB constructs in that particular study, attitudes toward the behavior significantly increased in the intervention group. Other interventions regarding nutritional behavior, fruit and vegetable consumption, and infant feeding showed significant improvement in all the TPB constructs, behavior change intentions, and self-reported safe food practice behaviors [49,50,51].

Even though all the constructs showed a significant improvement after the delivery of the intervention, three specific food safety practices were significantly improved in all the TPB constructs (attitudes toward the behavior, PBC, and subjective norms). These three practices were (1) chilling and storing procedures of a large pot of soup, (2) using a food thermometer to check the safe cooking temperature of ground beef, and (3) checking the temperature of the refrigerator and freezer (Table 4). There was a significant change in behavior intentions and self-reported safe food practice behaviors of the two practices related to thermometer use. All the food safety practices evaluated are in Appendix A.

The three specific practices (chilling and storing procedures, using a food thermometer, and checking refrigerator and freezer temperatures) that participants improved have been broadly studied among other researchers. A previous study assessing Hispanic consumers’ food safety knowledge showed that few participants knew the proper storage practices for a large pot of soup [52]. Barrett et al. [17] also found that consumers were not aware of how to chill large quantities of food properly. That study was conducted over three years. After a video intervention, participants from one of those years demonstrated significant improvement when asked about the proper way to store large quantities of food. The other practice that participants increased significantly was using a thermometer to check the safe cooking temperature of ground beef. Multiple research studies have shown that this is not a common practice among commercial food handlers nor household food handlers [53]. Other studies have demonstrated that consumers lack knowledge about the safe cooking temperature of different foods [46,54]. However, a food safety curriculum developed for high school students demonstrated that after students acquired certain food safety knowledge and skills, they perceived that a cooking thermometer was an appropriate method to check the safe cooking temperature of food, and some even reported using it more frequently [45]. The last behavior with a significant change was checking the temperature of the refrigerator and freezer. Consumers rarely check refrigerator and freezer temperatures at home and are still unaware of the safety consequences of an inadequate temperature [55]. Koidis et al. [56] found that 75% of consumers in their study did not know the recommended refrigerator temperature. Towns et al. [57] found that fewer than one-quarter of consumers in their study had a freezer and/or a refrigerator thermometer. Borrusso et al. [58] performed a visual audit of consumers’ home kitchens and reported that 43% of the refrigerators were not under the recommended temperature and only 4% of the refrigerators had a thermometer. However, educational interventions have been effective in making consumers aware of checking the correct refrigerator and freezer temperatures [59]. Organizations and agencies have also provided recommendations related to these practices. In the case of storing a large pot of soup, the Partnership for Food Safety Education recommends dividing large quantities of food into shallow containers so they can chill faster inside the refrigerator [60]. The U.S. Food and Drug Administration advises consumers to use cooking thermometers to check food doneness and also to keep their refrigerator and freezer at 4 °C and −18 °C, respectively [61].

After the individual analysis of the TPB constructs, behavior change intentions, and self-reported safe food practice behaviors, this study presents how the three constructs are associated with one another. Overall, the three independent constructs (attitudes toward the behavior, PBC, and subjective norms) of the TPB give an idea of the drivers toward the behavior change intentions, and, hence, the self-reported safe food practice behaviors. A Spearman correlation showed a significant relationship between the attitudes toward the behavior, PBC, subjective norms, the behavior change intentions, and the self-reported safe food practice behavior among participants (Figure 1). The two constructs with a significant correlation with the behavior change intentions and the self-reported safe food practice behaviors were the PBC and the subjective norms. As shown in Figure 1, both constructs observed a moderate positive correlation with the behavior change intentions while a weak positive correlation with the self-reported safe food practice behavior.

The three TPB constructs are independent predictors of the behavior change intentions. However, research that has included correlation analysis has demonstrated some weak and moderate associations among the constructs [62]. Consider, for example, an individual who is the primary meal preparer of a household with children and learns (forms the belief) that using a food thermometer to check meat doneness can decrease the risk of getting sick when consuming contaminated meat due to pathogenic bacteria. The new behavioral belief built based on the new information acquired could increase the attitude toward using a food thermometer to check meat doneness. However, if their peers and family members agree with this information, the individual will feel able to gain approval toward it, having a more supportive subjective norm; if the individual feels capable of using a food thermometer to check the meat doneness, the PBC will increase. This example illustrates how the different constructs can provide a certain correlation among each of them even though they are independent drivers for the behavior change intention.

In this study, the prediction of each of the individual constructs’ link to the behavior change intentions was not performed due to the small sample size and, hence, low power of the analysis [63]. Additionally, no rule of thumb was taken into consideration to perform the prediction (regression). However, some studies have predicted hand hygiene behavior of caterers [64] and farmworkers [65], finding that the PBC was a significant predictor for this specific behavior. In contrast, Lin and Roberts [66] found that subjective norms are more influential in predicting food safety behavior compared to the other TPB constructs. However, these researchers stated clearly that all constructs helped predict those behaviors but that one construct tended to have more influence (depending on the study). As previously mentioned, in the present study, only the correlation of all the TPB constructs, the behavior change intentions, and the self-reported safe food practice behaviors was evaluated. The PBC and subjective norms from this study have a moderate correlation with the behavior change intentions, but it can be observed that the PBC correlation coefficient is empirically greater than the subjective norms. This empirical finding suggests that the previous research examples align with our results. Both PBC and subjective norms have a significant association with the behavior change intentions of participants, but one tended to exert a greater influence. It is clear that a unique factor cannot explain human behavior, but multiple factors can lead to a better understanding of human behavior. This research demonstrated that the use of the TPB could strengthen food safety programs and trainings. Although this was not a longitudinal study to measure participants behavior on multiple occasions, a sign of the intention to perform food safety practices and, hence, actually perform those practices was shown.

### 3.3. Knowledge Assessment

A knowledge assessment was included in the pre- and post-survey to measure the level of understanding of the core four food safety practices and the impact of the food safety program. As shown in Table 3, participants’ overall food safety knowledge mean score (*p* < 0.001) significantly increased from 5.32 points (pre-survey) to 7.43 points (post-survey). Most knowledge items measured had a significant increase (*p* ≤ 0.05, Appendix A). Three items increased over 40% from the pre- to the post-survey (*p* ≤ 0.05), including the recommended way to store a large pot of soup (pre: 58%; post: 100%), recommended refrigerator temperature (pre: 52%; post: 93%), and the safe temperature for cooking ground beef (pre: 34%; post: 75%).

Knowledge has been the major component of evaluating interventions, regardless of the theories, formats, or target population, researchers have used in their interventions [59,67]. Different types of interventions can be found in the literature: some have evaluated courses, workshops, and campaigns, among others [45,68,69]. Some interventions and evaluations have included behavior change theories such as the TPB [26] and the Health Action Model [70]. All of these interventions evaluated participants’ knowledge finding food safety knowledge gaps. Nevertheless, after the interventions, participants had significantly increased their understanding of food safety practices. Little is known about food safety interventions for low-income individuals and the impact on their knowledge. However, some studies have assessed low-income individuals’ knowledge and found a gap around kitchen sanitation, recommended refrigerator temperature, and foods associated with *Listeria*, *Campylobacter*, and *Staphylococcus* [71,72]. Hence this current study is one of the few of its kind that has demonstrated that a food safety intervention for low-income individuals can also increase their knowledge regarding food safety practices.

### 3.4. Differences between English and Spanish participants

Even though English and Spanish participants all had a significant increase in their knowledge regarding the core four food safety practices, participants from the English (pre: 5.80 ± 1.32; post: 7.70 ± 0.75) program obtained higher mean scores than Spanish (pre: 4.83 ± 1.76; post: 7.17 ± 0.87) participants on the pre- and post-survey knowledge assessment questions (*p* < 0.001, Table 3). Food safety knowledge differences between English and Spanish participants have also been found in other previous studies. Panchal et al. [73] found a significant difference among English-speaking and Spanish-speaking participants when assessing the food safety knowledge from a group of food handlers in Chicago. Similar results were found in an interview conducted with restaurant managers and workers in which food safety-related knowledge was associated with participants’ primary language [74]. A survey conducted among the Hispanic population in Connecticut showed that only 5% of the 100 participants knew the definition of cross-contamination, and those who preferred to be interviewed in English were more likely to know such a definition [11]. Another factor that could cause the differences in the food safety knowledge increase was that the facilitators for the English and Spanish programs were different. Variance between facilitators’ food safety knowledge and their ability to explain certain concepts during the interaction with participants could result in the different learning outcomes. While other factors could also influence the difference among English and Spanish participants, such as the ethnicity or racial groups [75], comparisons cannot be established since inside the English program, two main racial groups were identified, while in the Spanish program there was just one major ethnic group. However, we do not reject the possibility that ethnicity and racial groups could also have and exert influence on food safety practices.

Cultural factors may also influence the social pressure that individuals perceive to conduct certain behaviors. For example, in the study by Bai et al. [76] about hygienic food handling behavior in a Chinese cultural context, the researchers replaced the subjective norms construct with face consciousness and conformity consciousness to align with the Chinese culture. They found that those constructs were significant predictors, and future interventions targeting that population should also include them. Face consciousness refers to how an individual tries to fit in to a specific society by increasing their reputation [77]. On the other hand, conformity consciousness refers to how an individual tries to replicate another person’s behavior [76]. Another example is found within the Hispanic population. Previous research has identified a characteristic of Hispanics residing in the USA, which is familism [78]. The American Psychological Association [79] defines familism as a cultural value in which interpersonal relationships with family members are over individual interests. These are only examples of how individuals in each culture can have unique characteristics that could be considered when targeting these populations for food safety training or educational programs.

### 3.5. Dialogue-Based Program Evaluation

As part of the evaluation of the food safety program’s impact, participants’ perception of the program was collected (Table 5). They reported that their program expectations were met (4.95 ± 0.29) and agreed that they would recommend this program to friends and family (4.92 ± 0.28). These results drive the researchers to suggest that the implementation of dialogue-based approaches was successful [29]. Vella [80] explained that dialogue education “protects” the learner from factors such as the teacher and the teaching methodology that could impact the learning process. They also commented that the dialogue-based approaches allowed the learner to build a more robust learning process by expressing their thoughts and engaging in their learning process. Other studies have also shown that learners in this educational environment develop skills regarding critical and analytical thinking. Smith and Haynes [81] showed that dialogue-based educational programs were effective for learners regarding topics related to criminal justice because they allowed them to analyze and understand different perspectives.

Dialogue-based methodology is rapidly increasing in many educational settings in which facilitators or teachers do not have to be present when the educational materials are being delivered. Some studies have a focus on intelligent dialogue-based methodology in which learners can interact with chatbots, and the chatbot can provide the answers or more information in a natural language [82]. This format adaptation is making education more tailored to individual learners and is reducing the cost of education. Afzal et al. [83] developed a dialogue-based facilitator chatbot and found that natural human language was one of the critical factors to the program’s success. The natural human language can be understood as the way people normally express their thoughts. Other researchers, Wambsganss et al. [84], created “ArgueTutor”, a conversational facilitator chatbot that provides feedback to students on their writing skills, and found that students demonstrated better argumentation skills in their written texts when using it. Even though the examples using this methodology are unrelated to food safety topics, they show promising outcomes for education and could be implemented in future virtual food safety educational programs.

### 3.6. Advantages of the Virtual Program for Low-Income Families

This manuscript has pointed out the effectiveness of a dialogue-based virtual food safety educational program for low-income families. However, other advantages of this virtual format were found in the process. The virtual modality of the program provided to the different type of learners (visual, auditory, and kinesthetic) an opportunity to acquire knowledge and practices related to food safety based on their learning preferences. For example, the pre-recorded slides used for the virtual program allow auditory learners to acquire knowledge related to food safety practices while listening to the material displayed by the program’s leading instructor. On the other hand, kinesthetic learners may find helpful the take-home tasks assigned after each session. Other studies in different research areas have also demonstrated the importance of designing educational materials to fit the different types of learners to increase audience engagement and achieve better results [85]. A case study from Clemons showed that it is important to analyze your target audience’s preferred learning styles to avoid developing and delivering a format that is suitable for the instructor but not for that specific audience [86]. These different learning styles used when developing the virtual food safety program for low-income families may explain the success in the effectiveness results. Additionally, the materials of this intervention were recorded and made available online for those who wanted to check back on the information to strengthen what they had learned in the sessions.

Another advantage of this virtual educational program was the languages it was delivered in, including English and Spanish. English is the most spoken language in the United States; however, the Hispanic population in the country has been increasing, and, hence, the Spanish language speakers [87]. A recent food safety education research found that delivering food safety training in different languages is more accessible in a virtual format since the availability of bilingual trainers with food safety expertise can be challenging to find only in one specific geographical area [88]. Moreover, participants receiving an educational program from instructors with similar backgrounds can be beneficial for the learning experience [89]. Additionally, providing information to the target audience in their first language can positively influence their learning process and reduce the wording gaps [90]. This current research study is in accordance with the study from Beary et al. [88] on the different opportunities that virtual food safety education can have, for example, the different learning modalities, the language, and the hybrid methodologies that these types of programs can offer to different target audiences.

Nowadays, the opportunity to develop and deliver information virtually is increasing as well as its accessibility. It cannot be generalized that all American populations have access to internet-enabled devices and internet services since some minority groups and areas in the US might have limited access [91]. However, policies have targeted this population to help them access internet-enabled devices and internet services, giving them more opportunities through this digital era [92]. The Federal Communications Commissions (FCC) has advocated through Comcast to develop a program for low-income individuals to have the “Internet Essentials [93]”. The 2018 report from the program “Internet Essentials” by Comcast showed that from 2012 through 2018, around 1.5 million low-income households (around 6 million individuals) benefited [94]. That report also showed that for most of the beneficiaries, it was the first time connecting to internet services. This suggests an increasing potential for virtual food safety programs targeting low-income families, as accessibility to different types of technologies is increasing among this population. Food safety educators need to continue using various resources available to close the knowledge and behavior gap regarding food safety.

## 4. Limitations and Future Directions

This study was planned and conducted carefully, despite some significant limitations. The study was performed at the beginning of the COVID-19 pandemic, which made the recruitment of program participants difficult and impeded the ability to reach the desired sample size. Due to the resultant small sample size obtained per group (English-speaking and Spanish-speaking participants), data analysis was limited to mean comparisons and correlation analysis. The researchers who conducted this study acknowledge that the findings cannot be generalized to a broader low-income population worldwide because this population could experience varying difficulties that are unique to individual countries. However, the findings provide a strong path for future food safety programs for low-income populations in the United States. The evaluation of this food safety program also serves as a starting point for food safety educators from other nations to explore alternative ways to deliver food safety education for their populations.

However, in the future, to investigate more variables and identify different outcomes of the program, a larger sample size should be collected. With a larger sample size, researchers could explore the effects of the sociodemographic variables on the change in knowledge, TPB constructs, behavior change intentions, and self-reported behaviors. Additionally, the TPB methodology, in combination with the use of regression models, may be useful in predicting the behavior change intentions of the program participants.

Limitations in the time availability of participants made scheduling the sessions for this study difficult. Moreover, not all of the sessions had the same number of participants, which could influence participants’ dialogue-based learning experience, and, thus, their behavior change intentions. For this study, the English-speaking participants were divided mainly among White and African Americans, but under ideal circumstances ethnicity and racial groups should be taken as different groups because they may experience differing barriers to the adoption of food safety practices.

Multiple factors can contribute to the results gathered when evaluating each of the TPB constructs, such as the sample size used, the demographic characteristics of participants, when participants are evaluated, and how the intervention and evaluation tools were developed [62,95]. The TPB is very specific to the behaviors being evaluated and the target population. Previous studies using the TPB suggested implementing an extended model because participants’ background and cultural factors can influence their intentions to adopt food safety practices [64,76,96]. However, before implementing any new construct inside the TPB model, a literature review should be conducted to corroborate which factors can influence participants’ behavior change intentions, and to then incorporate them in the intervention and evaluation. Observational studies could support the TPB model’s accuracy.

Regarding the program evaluation, no follow-up evaluation was carried out in this study to assess participants’ knowledge retention for a specified period of time after completing the program. Future research should include longitudinal measures to evaluate participants’ knowledge retention and their level of engagement in recommended food safety practices after the program.

## 5. Conclusions

This study evaluated a virtual dialogue-based food safety intervention using the theory of planned behavior (TPB). The results revealed that the TPB helped to discern the behavior change intentions of English- and Spanish-speaking low-income participants regarding certain food safety practices. The food safety intervention was able to significantly increase participants’ overall knowledge, attitudes toward the behavior, PBC, subjective norms, and behavior change intentions. In addition, our findings revealed some differences in knowledge gain among English- and Spanish-speaking low-income individuals. Even though a significant improvement was demonstrated in both groups, English participants had a higher knowledge score than Spanish participants.

The study findings provide a deeper understanding of food safety practices in low-income families and offer unique perspectives on how virtual instruction and dialogue presentations can help disseminate food safety information effectively. This program can potentially reach and benefit a larger population if offered through the Supplemental Nutrition Assistance Program (SNAP), School Pantry Program, Mobile Pantry, and other food assistance programs [97,98,99].

The virtual program is highly adaptable and can be translated to meet the needs of specific populations, especially those niche groups. Even beyond the COVID-19 pandemic, virtual materials and platforms will remain a viable resource to reach and connect with these and other underrepresented groups. Food safety educators need to be prepared to use these tools to advance food safety educations within these groups.

## Figures and Tables

**Figure 1 foods-11-00355-f001:**
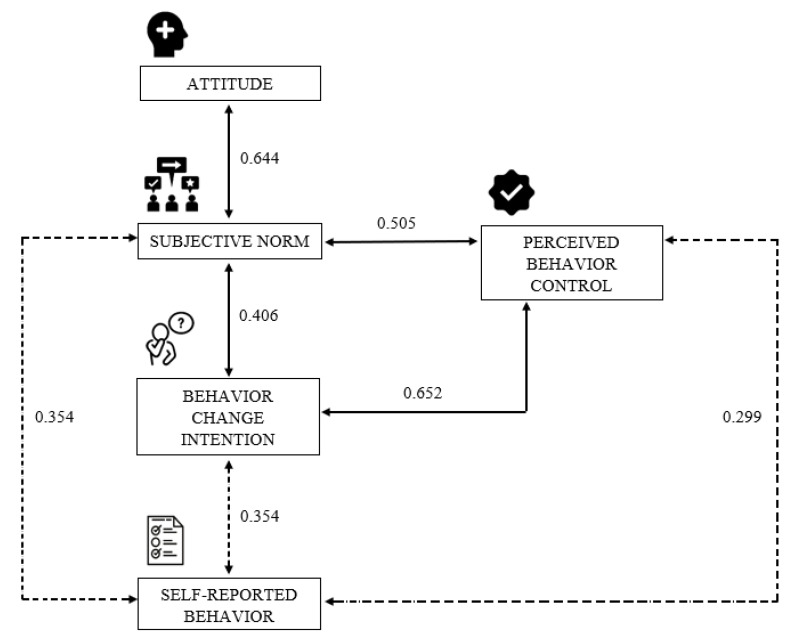
Spearman correlation between the theory of planned behavior constructs, behavior change intentions, and self-reported safe food practice behaviors after the intervention.

**Table 1 foods-11-00355-t001:** Spearman correlation coefficient’s interpretation.

Correlation Coefficient		Interpretation
+1	−1	Perfect
+0.9	−0.9	Strong
+0.8	−0.8	Strong
+0.7	−0.7	Strong
+0.6	−0.6	Moderate
+0.5	−0.5	Moderate
+0.4	−0.4	Moderate
+0.3	−0.3	Weak
+0.2	−0.2	Weak
+0.1	−0.1	Weak
0	0	Zero

This table was adapted from Akoglu, “User’s guide to correlation coefficients”.

**Table 2 foods-11-00355-t002:** Sociodemographic characteristics of the virtual food safety educational program.

Characteristics	English Program *n* (%) (*n* = 30)	Spanish Program *n* (%) (*n* = 30)	Total *n* (%) (*n* = 60)
Gender			
Female	30 (100)	28 (93)	58 (97)
Male	-	2 (7)	2 (3)
Prefer not to answer	-	-	-
Age			
18–24	1 (3)	5 (17)	6 (10)
25–34	16 (54)	8 (27)	24 (40)
35–54	12 (40)	16 (53)	28 (47)
55 and above	1 (3)	1 (3)	2 (3)
Prefer not to answer	-	-	-
Ethnicity			
White	14 (47)	2 (7)	16 (26)
Hispanic American	1 (3)	23 (77)	24 (40)
Asian American	1 (3)	-	1 (2)
Native American	1 (3)	-	1 (2)
African American	10 (34)	-	10 (17)
Others	3 (10)	5 (16)	8 (13)
Children (<10 years old) in household			
1	5 (17)	11 (37)	16 (26)
2	10 (33)	9 (30)	19 (32)
3	11 (37)	2 (7)	13 (22)
4	4 (13)	1 (3)	5 (9)
5	-	-	-
6 and above	-	-	-
Prefer not to answer	-	7 (23)	7 (11)
Meal preparation frequency			
All the time	12 (40)	16 (53)	28 (47)
Nearly all the time	16 (53)	9 (30)	25 (42)
Some of the time	2 (7)	5 (17)	7 (11)
Never	-	-	-
Household income			
Less than 10,000USD	3 (10)	3 (10)	6 (10)
10,001USD–30,000USD	8 (27)	8 (27)	16 (27)
30,001USD–50,000USD	16 (53)	4 (13)	20 (34)
50,001USD–80,000USD	3 (10)	5 (16)	8 (13)
80,001USD and above	-	2 (7) ^a^	2 (3)
Prefer not to answer	-	8 (27)	8 (13)
Education			
High school/GED degree	15 (50)	6 (20)	21 (35)
Associate degree	4 (13)	9 (30)	13 (22)
Bachelor’s degree	8 (27)	14 (47)	22 (36)
Post-graduate degree	3 (10)	1 (3)	4 (7)

^a^ These individuals are classified as low-income based on the U.S. Department of Housing and Urban Development’s 2019 Section 8 income limit values (https://www.huduser.gov/portal/datasets/il/il19/Section8-IncomeLimits-FY19.pdf [accessed on 1 April 2021]). The household income limits are calculated between the median family incomes and Fair Market Rents, including the size and location of the family as part of the influential factors of the calculation.

**Table 3 foods-11-00355-t003:** Overall measurements of knowledge, theory of planned behavior (TPB) constructs, behavior change intentions, and self-reported safe food practice behavior before and after the intervention.

	English Program (*n* = 30)			Spanish Program (*n* = 30)			Total (*n* = 60)		
	Pre-Survey (Mean ± SD)	Post-Survey (Mean ± SD)	*p* Value (Paired Samples *t*-Test)	Pre-Survey (Mean ± SD)	Post-Survey (Mean ± SD)	*p* Value (Paired Samples *t*-Test)	Pre-Survey (Mean ± SD)	Post-Survey (Mean ± SD)	*p* Value (paired Samples *t*-Test)
Knowledge ^a^	5.80 ± 1.32	7.70 ± 0.75	<0.001	4.83 ± 1.76 ^d^	7.17 ± 0.87 ^e^	<0.001	5.32 ± 1.62	7.43 ± 0.85	<0.001
Attitudes toward the behavior ^b^	30.50 ± 2.61	33.03 ± 1.75	<0.001	29.73 ± 3.83	33.00 ± 1.98	<0.001	30.12 ± 3.27	33.02 ± 1.85	<0.001
Perceived behavioral control (PBC) ^b^	31.67 ± 2.45	34.47 ± 1.31	<0.001	31.57 ± 3.78	34.37 ± 1.19	<0.001	31.62 ± 3.16	34.42 ± 1.24	<0.001
Subjective norms ^b^	31.90 ± 4.59	35.10 ± 4.21	<0.001	34.53 ± 4.15 ^d^	36.00 ± 3.64	0.086	33.22 ± 4.54	35.55 ± 3.93	<0.001
Behavior change intentions ^b^	27.73 ± 1.74	29.67 ± 0.84	<0.001	28.13 ± 2.06	29.70 ± 0.60	<0.001	27.93 ± 1.90	29.68 ± 0.72	<0.001
Self-reported safe food practice behaviors ^c^	25.80 ± 4.44	30.17 ± 4.07	<0.001	23.77 ± 3.53 ^d^	28.43 ± 4.10	<0.001	24.78 ± 4.10	29.30 ± 4.14	<0.001

^a^ The mean score of knowledge is based on a total score of 8. ^b^ The mean scores of the attitudes toward the behavior, perceived behavior control, and self-reported behaviors are based on a total score of 35. ^c^ The mean score of the subjective norms are based on a total score of 40. The mean score of the behavior change intentions are based on a total score of 30. ^d^ The mean score between the English program and Spanish program pre-survey was significantly different, *p* ≤ 0.05. ^e^ The mean score between the English program and Spanish program post-survey was significantly different, *p* ≤ 0.05.

**Table 4 foods-11-00355-t004:** Food safety practices with a significant improvement in the TPB constructs, behavior change intentions, and self-reported safe food practice behaviors.

Practice	TPB Constructs			Behavior Change Intentions	Self-Reported Safe Food Practice Behaviors
	Attitudes toward the Behavior	PBC	Subjective Norms		
Chilling and storing procedures for a large pot of soup. ^a^	pre: 4.17 ± 0.91 post: 4.90 ± 0.35 *p* value: ≤0.05	pre: 4.38 ± 0.76 post: 4.88 ± 0.37 *p* value: ≤0.05	pre: 3.98 ± 1.10 post: 4.48 ± 0.77 *p* value: ≤0.05	NA ^b^	NA ^b^
Using thermometer to check the safe cooking temperature of ground beef. ^a^	pre: 3.92 ± 0.98 post: 4.87 ± 0.34 *p* value: ≤0.05	pre: 4.13 ± 1.00 post: 4.88 ± 0.37 *p* value: ≤0.05	pre: 3.38 ± 1.35 post: 4.32 ± 0.83 *p* value: ≤0.05	pre: 4.07 ± 0.97 post: 4.92 ± 0.28 *p* value: ≤0.05	pre: 2.20 ± 1.39 post: 3.93 ± 1.12 *p* value: ≤0.05
Checking the temperature of the refrigerator and freezer. ^a^	pre: 4.23 ± 1.16 post: 4.93 ± 0.31 *p* value: ≤0.05	pre: 4.37 ± 0.80 post: 4.92 ± 0.28 *p* value: ≤0.05	pre: 4.12 ± 0.99 post: 4.55 ± 0.70 *p* value: ≤0.05	pre: 4.53 ± 0.60 post: 4.97 ± 0.18 *p* value: ≤0.05	pre: 2.65 ± 1.54 post: 4.17 ± 1.26 *p* value: ≤0.05

^a^ For the specific statement used in the TPB constructs, behavior change intentions, and self-reported safe food practice behaviors, refer to Appendix A. ^b^ The statement was not presented to participants for the behavior change intentions and the self-reported safe food practices.

**Table 5 foods-11-00355-t005:** Participant evaluation of the virtual food safety educational program.

	English Program (*n* = 30)	Spanish Program (*n* = 30)	Total (*n* = 60)
Statements ^a^	Evaluation (Mean ± SD)		
My expectations were met.	5.00 ± 0.00	4.90 ± 0.40	4.95 ± 0.29
I would recommend this course to my friends and family.	5.00 ± 0.00	4.83 ± 0.38	4.92 ± 0.28
I have practiced what I have learned in my daily food preparation routine.	4.90 ± 0.31	4.97 ± 0.18	4.93 ± 0.25
This course will have a significant impact on the safety of my food handling practices.	4.93 ± 0.25	4.83 ± 0.46	4.88 ± 0.37

^a^ The statements were measured on a 5-point Likert scale: 1, strongly disagree; 5, strongly agree.

## Data Availability

Not applicable.

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
