# Peer review of "An Evaluation of a Virtual Food Safety Program for Low-Income Families: Applying the Theory of Planned Behavior"

_foods, 2022, doi:10.3390/foods11030355_

Round 1

Reviewer 1 Report

dear authors

congratulations on a wonderful and interesting research paper. following points need consideration:

  1. how was the sample size collected
  2. the objective was to see change before and after intervnetion. a subgroup comparison between English and Spanish speaking subjects is not recommended as the sample size is too small for generalisability of finding wrt change in knowledge score 
  3. there were only 3 males and the remaining respondents were males. is it possible to remove the two subjects from analysis 
  4. the authors are advised to consider logistic regression to see effect of variables like age, education, number of children with change in scores
  5. the spearman coeff was of moderate values (<0.6) and were is the correlations and hence the same may not be overemphasized in the discussion.

Author Response

Answers to The Review Report (Round 1)

Reviewer 1

Comment 1: how was the sample size collected

Comment 2: the objective was to see change before and after intervnetion. a subgroup comparison between English and Spanish speaking subjects is not recommended as the sample size is too small for generalisability of finding wrt change in knowledge score

Response: We want to thank the reviewer for pointing out an important component of this study. The response is based on our interpretation of the comments and questions. In this response, we will be addressing comments 1 and 2 simultaneously.

We understand any concerns that you may have regarding type I and II errors when analyzing the data due to the sample size. The overall sample was N = 60, and for each group (English-speaking and Spanish-speaking), the sample size was N = 30. To determine our study’s sample size, we referred to previous evaluations of food safety interventions; some examples are provided in Table 1. Also, we searched if the sample size selected was appropriate for comparisons. King and Eckersley (2019), as well as Pathack (2011), comment that a sample size of N ≤ 30 is appropriate for the Student’s t-test, and the justification relies on the t-distribution of the data (Table 2). The Student’s t-test was originally developed for small sample sizes; however, it can also be used when having a larger sample size.

A sentence in the manuscript in lines 187-189 was included to clarify the sample size used for the study.

“In addition, the sample size for this study was determined based on previous food safety intervention programs [26,36,37], and an appropriate number of participants needed to perform statistical analysis of pre- and post-data collected [38,39].”

Table 1. Example of studies evaluating food safety interventions

Authors

Year

Title

Intervention and Sample Size (N)

Milton and Mullan

2012

An application of the theory of planned behavior – A randomized controlled food safety pilot intervention for young adults

a.     Intervention (15)

b.     General control (15)

c.     Mere measurement control (15)

Smith et al.

2014

Does food safety training for non-profit food service volunteers improve food safety knowledge and behavior?

a.     Food safety training was given by a ServSafe® Certified instructor (34)

Gold et al.

2014

Discussion map and cooking classes: Testing the effectiveness of teaching food safety to immigrants and refugees

a.     Food safety map (27)

b.     Cooking class (21)

c.     Control (25)

Table 2. Books’ chapter discussing the Student’s t-test

Authors

Chapter’s title

Year

Book’s title

Pages

King and Eckersley

Chapter 5 - inferential statistics II: parametric hypothesis testing

2019

Statistics for biomedical engineers and scientists: how to visualize and analyze data

91-117

Pathak

Chapter 5. significance of difference of means

2011

Statistics in Education and Psychology

NAa

aOnline version is available, no page number is provided.

Comment 3: there were only 3 males and the remaining respondents were males. is it possible to remove the two subjects from analysis

Response: We want to thank the reviewer for the suggestions. The response is based on our interpretation of the suggestion.

We understand any concerns of having only two males (3%) in our sample. However, we consider that if we delete those two males from this research, we will be removing an important approximation on how females remain the primary meal preparers at their household inside this niche population. Jarlais et al. (2004) point out that excluding data from research related to public health could bias the findings of the study. In addition, those two males were included in this study based on the inclusion criteria. Osborne (2004) provides an example of what could be a sampling error when the subjects are from a different population. This researcher points out that in a nurse study referring to floor nurses, it would be inappropriate to include a nurse that was moved into a hospital administration position.

Table 3. Articles presented in response to comment 3

Authors

Year

Title

Osborne

2004

The power of outlier (and why researcher should always check for them)

Jarlais et al.

2004

Improving the reporting quality of nonrandomized evaluations of behavioral and public health interventions: The TREND Statement

Comment 4: the authors are advised to consider logistic regression to see effect of variables like age, education, number of children with change in scores

Response: We want to thank the reviewer for the suggestion. The response is based on our interpretation of the suggestion.

It will be interesting to see how some sociodemographic variables could affect the change in scores and who is more likely to have that change. However, our main concern using logistic regression is that some internal sociodemographic groups may only have one to two participants. According to Nemes et al. (2009) and van Smeden et al. (2019), the small number of subjects among each group being compared could decrease the power of the analysis and may overestimate any significant finding.

A sentence in the manuscript in lines 548-550 was included to show the limitation of the sample size.

“Due to the resultant small sample size obtained per group (English-speaking and Spanish-speaking participants), data analysis was limited to mean comparisons and correlation analysis.”

A sentence in the manuscript in lines 557-562 was included to show future data analysis directions with a larger sample size.

“But in the future, to investigate more variables and identify different outcomes of the program, a larger sample size should be collected. With a larger sample size, researchers could explore the effects of the sociodemographic variables on the change in knowledge, TPB constructs, behavior change intentions, and self-reported behaviors. Also, the TPB methodology, in combination with the use of regression models, may be useful in predicting the behavior change intentions of the program participants.”

Table 4. Articles presented in response to comment 4

Authors

Year

Title

Nemes et al.

2009

Bias in odds ratios by logistic regression modelling and sample size

Van Smeden et al.

2019

Sample size for binary logistic prediction models: beyond events per variable criteria

Comment 5: the spearman coeff was of moderate values (<0.6) and were is the correlations and hence the same may not be overemphasized in the discussion.

Response: We want to thank the reviewer for the comment. The response is based on our interpretation of the comment.

We understand your concern. However, we used certain words to avoid overemphasizing the findings.

The words “this empirical finding” were used in lines 393-394.

“This empirical finding suggests that the previous research examples align with our results.”

The words “could strengthen” were used in lines 397-398.

“This research demonstrated that the use of TPB could strengthened food safety programs and trainings.”

The words “a proximity on the intention to perform” were used in lines 399-401.

“Although this was not a longitudinal study to measure participants behavior in multiple occasions, a proximity on the intention to perform food safety practices and hence perform those practices was shown.”

Reviewer 2 Report

The authors explore an interesting topic for Foods’ readers such as the effect of virtual food safety educational program targeting food handlers in low-income families using a sample of 60 recruited individuals in the U.S.. Authors found that virtual educational program improved low-income individuals’ food safety knowledge that changed their food safety attitudes and behaviors. The paper is well written and properly structured as well as the statistical analysis has been properly performed.

From my perspective, the paper would benefit from the inclusion of a literature review section in which discuss studies investigating on food safety, food insecurity and food access referring to the U.S. case study. Here some studies to which authors may look at:

i) Bonanno, A., & Li, J. (2015). Food insecurity and food access in US metropolitan areas. Applied Economic Perspectives and Policy37(2), 177-204.

ii) Sharpe, P. A., Liese, A. D., Bell, B. A., Wilcox, S., Hutto, B. E., & Stucker, J. (2018). Household food security and use of community food sources and food assistance programs among food shoppers in neighborhoods of low income and low food access. Journal of hunger & environmental nutrition13(4), 482-496.

iii) Seligman, H. K., & Berkowitz, S. A. (2019). Aligning programs and policies to support food security and public health goals in the United States. Annual review of public health40, 319-337.

The inclusion of a literature review can help scholars to provide more emphasis on the novelty of their work compared to existing ones as it is not clear enough the genuine contribution of the current work compared to extant studies in the literature.

Also, a large part of the conclusions is not appropriate, as it is only a concise repetition of the comments. In the conclusions, you should simultaneously consider all you have discovered, and exploit it to add something new (or new interpretations), and policy indications.

The authors do not discuss possible limitations of their study or the insights for future directions of research. Maybe they could discuss external validity of the results in terms of possible insights in other countries and/or additional variables that they would have liked to have to better answer to their research question.

Author Response

Answers to The Review Report (Round 1)

Reviewer 2

Comment 1: From my perspective, the paper would benefit from the inclusion of a literature review section in which discuss studies investigating on food safety, food insecurity and food access referring to the U.S. case study. Here some studies to which authors may look at:

  1. i) Bonanno, A., & Li, J. (2015). Food insecurity and food access in US metropolitan areas. Applied Economic Perspectives and Policy, 37(2), 177-204.
  2. ii) Sharpe, P. A., Liese, A. D., Bell, B. A., Wilcox, S., Hutto, B. E., & Stucker, J. (2018). Household food security and use of community food sources and food assistance programs among food shoppers in neighborhoods of low income and low food access. Journal of hunger & environmental nutrition, 13(4), 482-496.

iii) Seligman, H. K., & Berkowitz, S. A. (2019). Aligning programs and policies to support food security and public health goals in the United States. Annual review of public health, 40, 319-337.

Response: We want to thank the reviewer for the suggestion. The response is based on our interpretation of the suggestion.

We consider that it will be interesting to conduct a literature review investigating food safety, food insecurity, and food access. However, these topics may be out of the scope of this study. But your comment is valuable, and the references provided were included in the manuscript.

A sentence in the manuscript in lines 599-602 was included to link food security programs with the food safety program we developed and evaluated.

“This program can potentially reach and benefit a larger population if offered through the Supplemental Nutrition Assistance Program (SNAP), School Pantry Program, Mobile Pantry, and other food assistance programs [97-99].”

Comment 2: Also, a large part of the conclusions is not appropriate, as it is only a concise repetition of the comments. In the conclusions, you should simultaneously consider all you have discovered, and exploit it to add something new (or new interpretations), and policy indications.

Response: We understand the concern of the reviewer. The response is based on our interpretation of the concern.

We edited the limitations and future directions, as well as the conclusion to highlight the significance of the study. The significance of the study is summarized in Table 1.

Table 1. Summary of the significance of the study

Significance

Text

Lines

Other countries

“The evaluation of this food safety program also serves as a starting point for food safety educators from other nations to explore alternative ways to deliver food safety education for their populations.”

554-556

Virtual instruction and dialogue presentations for low-income individuals

“The study findings provide a deeper understanding of food safety practices in low-income families and offer unique perspectives on how virtual instruction and dialogue presentations can help disseminate food safety information effectively.”

597-599

Food safety virtual materials

“Even beyond the COVID-19 pandemic, virtual materials and platforms will remain a viable resource to reach and connect with these and other underrepresented groups. Food safety educators need to be prepared to use these tools to advance food safety educations within these groups.”

604-607

Comment 3: The authors do not discuss possible limitations of their study or the insights for future directions of research. Maybe they could discuss external validity of the results in terms of possible insights in other countries and/or additional variables that they would have liked to have to better answer to their research question.

Response: We thank the reviewer for the comment. The response is based on our interpretation of the comment.

We edited the limitations and future directions of the manuscript to provide more details. The limitations and future directions are summarized in Table 2 and 3, respectively.

Table 2. Summary of the limitations of the study

Limitations

Text

Lines

Sample size

“Due to the resultant small sample size obtained per group (English-speaking and Spanish-speaking participants), data analysis was limited to mean comparisons and correlation analysis.”

548-550

Generalization

“The researchers who conducted this study acknowledge that the findings cannot be generalized to a broader low-income population worldwide, because this population could experience varying difficulties that are unique to individual countries.”

550-552

Participants’ availability

“Limitations in the time availability of participants made scheduling the sessions for this study difficult. Also, not all the sessions had the same number of participants, which could influence participants’ dialogue-based learning experience, and thus their behavior change intentions.”

563-566

Ethnicity and racial groups

“For this study the English-speaking participants were divided mainly among White and African Americans, but under ideal circumstances ethnicity and racial groups should be taken as different groups because they may experience differing barriers to adoption of food safety practices.”

566-569

Table 3. Summary of the future directions of the study

Future directions

Text

Lines

Sample size and data analysis

“But in the future, to investigate more variables and identify different outcomes of the program, a larger sample size should be collected. With a larger sample size, researchers could explore the effects of the sociodemographic variables on the change in knowledge, TPB constructs, behavior change intentions, and self-reported behaviors. Also, the TPB methodology, in combination with the use of regression models, may be useful in predicting the behavior change intentions of the program participants.”

557-562

Theory of Planned Behavior

“Previous studies using the TPB suggested implementing an extended model because participants’ background and cultural factors can influence their intentions to adopt food safety practices [64,76,96]. However, before implementing any new construct inside the TPB model, a literature review should be conducted to corroborate which factors can influence participants’ behavior change intentions, and to then incorporate them in the intervention and evaluation.”

574-579

Observational study

“Observational studies could support the TPB model’s accuracy.”

579-580

Longitudinal study

“Future research should include longitudinal measures to evaluate participants’ knowledge retention and their level of engagement in recommended food safety practices after the program.”

583-585
